# LUMINA: LONG-HORIZON UNDERSTANDING FOR MULTI-TURN INTERACTIVE AGENTS

## ABSTRACT

Language models can excel at a variety of tasks (e.g., mathematical reasoning and coding) which are fundamental to solving more general goal-oriented feedback-driven agentic problems. However, based on recent findings, two key points are evident: (a) agentic problems require a variety of skills such as long-context reasoning, planning and decision making, and efficient exploration; (b) even large frontier models under-perform in these family of tasks, especially in problems requiring long-horizon understanding. For example, GPT-4o has a $48.8\%$ success rate on the AppWorld benchmark. In this paper, our goal is to understand the relation between the two, by examining which skills are necessary for solving multi-turn problems. We work towards this goal using an oracle counter-factual framework that allows us to answer the question: what if the agent could leverage a specific oracle skill to achieve its goal? To enable this framework, we introduce a set of procedurally-generated game-like tasks whose complexity can be controlled. For these controlled environments, we can provide accurate oracle interventions to guide the agent towards the goal. Our findings suggest that while most interventions (e.g., planning) are generally beneficial, for some interventions the utility depends on the intricacies of the benchmark (e.g., ability to track state while iteratively modifying python lists).

## 1 INTRODUCTION

Large Language Models (LLMs) have demonstrated exceptional performance across a wide range of tasks, including natural conversations (Touvron et al., 2023; Achiam et al., 2023), question answering (Yang et al., 2018), competitive coding (Austin et al., 2021; White et al., 2024), and mathematical reasoning (Comanici et al., 2025; Guo et al., 2025). Consequently, given their general-purpose capabilities, a natural question that is actively being explored is whether language models can be leveraged as *multi-turn* agents, i.e., whether they can iteratively perceive, reason, and take strategic actions towards achieving a distant goal. Such tasks introduce a host of new challenges, such as maintaining coherence over multiple turns and long contexts, reasoning over multiple dynamic pathways, recovering from errors, identifying the right tools for the task, and efficiently tracking state and progress without explicit feedback.

How capable are current language models as agents? To answer this, the community is actively working on numerous benchmarks to quantitatively analyze multi-turn agent capabilities across various domains, ranging from function calling (Patil et al., 2025), web navigation (Koh et al., 2024), interactive coding (Trivedi et al., 2024), human interaction (Liu et al., 2023), and game-playing (Guertler et al., 2025). Ongoing results suggest that there is plenty of progress to be made, e.g., on Appworld (Trivedi et al., 2024), GPT-4o has a success rate of 48.8% and open-weight models such as LLama3-70B (Grattafiori et al., 2024) achieve 24.4%. As many of the benchmark analysis reports, pushing progress requires enabling numerous skills, such as efficient task decomposition, planning, state tracking, and information gathering. However, it is largely unclear on which of these skills (or combination thereof) are the bottleneck to make progress towards capable multi-turn agents.

In this paper, our goal is to critically examine and understand skills that enable progress towards general-purpose long-horizon agents. To help us understand, we propose an oracle intervention framework that helps us evaluate the importance of skills by asking counterfactual questions. The framework helps us gauge agent's performance improvement when assisted by a skill-specific or-

acle or a combination of multiple skills. Using this framework, we investigate three oracle interventions: planning, tracking belief state, and context reformulation. However, constructing oracle interventions on real-world benchmarks (which typically involve human annotation and verification) is cumbersome and not straight-forward. Since multi-turn tasks admit numerous valid dynamic paths towards reaching the goal, annotating oracle solutions becomes intractable. To make oracle interventions tractable, we additionally propose a set of procedurally-generated game-playing environments where optimal actions and strategies can be reliably computed at any step of the roll-out. Specifically, we consider three environments: `ListWorld` (to evaluate multi-turn list modification capabilities), `TreeWorld` (to evaluate multi-turn graph traversal), and `GridWorld` (to evaluate 2D spatial navigation). All the environments are configurable, and importantly, enable us to inject accurate oracle information at any point of the agent's trajectory.

Our framework enables us to examine multi-turn agents at long-horizon tasks along multiple dimensions (e.g., task complexity, model size, influence of specific oracle skills) and helps us provide a number of insights. First, we observe a significant discrepancy with the low success rates over long-horizon trajectories in spite of high accuracy per-step (i.e., whether action is one of the optimal actions). For example, in tree search problems, we observe very low success rates ($<10\%$ in this case) despite high per-step accuracy ($>80\%$). This indicates that a dominant factor in difficulty of multi-turn environments is the fact that success in the task requires many correct steps, and even a small probability of error in each step prohibitively hinders task success. Apart from compounding errors (Sinha et al., 2025; Li et al., 2025), we also attribute making terminal errors (e.g., premature termination) contributing to the discrepancy. Second, we can leverage oracle interventions to understand contributions of specific skills that best contribute to improving success over multi-turn tasks. Here, we find that although oracle interventions generally help improve success rates, the degree to which they help significantly depends on other factors. For instance, while optimally pruning context (containing action-observation interaction history) helps smaller models ($\leq$8B parameters), it also shows to be counter-productive for larger models. Another factor that determines improvements is unsurprisingly the task itself: tasks that rely on accurately tracking belief state (e.g., those involving tracking hidden state) benefit the most from relevant state-tracking skills, while other tasks such as spatial navigation benefit heavily from planning. Overall, our findings present a double-edged picture: while improving specific skills (enabled by oracles interventions in our case) generally help the LLM-based agents in multi-turn, fully bridging the gap likely requires exploiting environment and model-specific understanding.

## 2 RELATED WORKS

**LLMs and Agents** Agent-based systems have a long history (Russell et al., 1995) and can be defined by an agent (the policy) perceiving and interacting with an environment towards achieving a goal and, in turn, receiving a reward. Recent literature demonstrates that capable language models can serve many roles within such systems, such as modeling the policy (Huang et al., 2022; Yao et al., 2023b), the environment as a world model (Hao et al., 2023), or the reward (Zheng et al., 2023; Zhang et al., 2025). We specifically focus on leveraging the LLM as a policy, which based on trajectory auto-regressively samples the next action. A notable and representative example is ReAct-based prompting (Yao et al., 2023b), which interleaves chain-of-thought thinking and taking task-specific actions at each step. ReAct prompting has been shown to be highly successful in a variety of domains, ranging from playing games (Wang et al., 2023a) to interactive coding agents (Trivedi et al., 2024). In this work, we use ReAct-based prompting to elicit dynamic reasoning and planning behavior from an LLM.

**LLM Agent capabilities** What makes for a good Language model agent? While models need to be fundamentally capable of language understanding and complex reasoning, a number of skills are required beyond this. The policy interacts with the environment with a set of admissible actions (e.g., tools, function calls) and hence need to be capable of calling functions (Qin et al., 2023; Patil et al., 2024) with appropriate arguments. Since tasks admit multiple paths towards the goal, the agents need to also be capable of multi-path reasoning (Besta et al., 2024; Yao et al., 2023a) and re-planning (Song et al., 2023) to revise actions in light of dynamic environmental feedback. Since decision making involves reflecting short-term (e.g., episode history in context window) and long-term (e.g., external storage) memory (Song et al., 2023; Huang et al., 2023; Wang et al., 2023b)

also plays an important role. The sequence of actions taken by the agent also modify the (hidden) state in the environment, and the models need to be adept in tracking its state (Ebrahimi et al., 2024; Vodrahalli et al., 2024). As enumerated above, a number of capabilities appear to play a crucial role towards enabling agentic use-cases.

**Characterizing Bottlenecks for Agents**    How can we assess the effectiveness of models in agentic tasks? One approach is to conduct holistic evaluations, while another is to analyze the contribution of specific capability dimensions, such as long-context reasoning, to marginal performance gains. Towards a holistic evaluation, a number of benchmarks exist (Trivedi et al., 2024; Patil et al., 2024) to evaluate an agent's capabilities. These benchmarks generally show a common trend: large models tend to significantly outperform smaller models. This motivates the question in our work: what is the bottleneck that leads to the performance discrepancies. Towards understanding this, a few works exist on characterizing bottlenecks of capabilities of language models over multi-turn tasks. Abdulhai et al. (2023) work towards this goal by understanding the influence of RL algorithms in strategic game-playing tasks (e.g., maze). Cemri et al. (2025) study error taxonomy of multi-agent systems across a range of popular benchmarks, such as AppWorld (Trivedi et al., 2024). Concurrent to our work, Sinha et al. (2025) study bottlenecks by isolating execution capabilities in long-horizon tasks. Similar to prior works, our goal too is to characterize bottlenecks of multi-turn agents by isolating capabilities. However, in contrast, we study the bottlenecks in procedurally-generated game-playing environments, which allows us to enable oracle interventions.

## 3    Formulation: LUMINA

In this section, we begin by detailing the underlying process that requires an agent to perform sequential decision-making to complete the task. To better enable the agent complete the task, we then elaborate on how to augment information at each turn using oracle interventions.

**POMDP Tasks**    We study tasks that can be modeled as a Partially-observable Markov Decision Process: $\mathcal{M} = \langle \mathcal{S}, \mathcal{A}, \mathcal{O}, T, \Omega, H, S_{\text{Goal}} \rangle$ where $\mathcal{S}$ is the hidden state space, $\mathcal{A}$ the agent's action space, and $\mathcal{O}$ is the observation space. Furthermore, $T : \mathcal{S} \times \mathcal{A} \to \mathcal{S}$ is the transition function (deterministic in our case) and $\Omega : \mathcal{S} \times \mathcal{A} \to \mathcal{O}$ is the observation function. The termination can be performed either by the agent (e.g., DONE action) or by the environment (e.g., $t \geq$ horizon $H$). For simplicity, we consider terminal reward function: the agent receives reward of $1$ if it terminates at the goal state $S_{\text{Goal}}$ within at most $H$ steps. The objective of the agent is to maximize the probability of success, which we refer to as the *success rate* and denote by $J(\pi_\theta)$ for an agent $\pi_\theta$.

**Base (ReAct) Policy Agent**    Towards taking sequential decisions to solve the task (represented as text $x$, we consider a stochastic policy modeled by a ReAct (Yao et al., 2023b) LLM agent $\pi_\theta$ where at each step $t$:

$$a_t \sim \pi_\theta \left( \cdot \mid x, h_{t-1} \right).$$

Here, $h_{t-1} := (a_1, o_1, ..., a_{t-1}, o_{t-1})$ is the history of the past interactions between the agent and the environment. Action $a_t$ consists of both a chain-of-thought text (which is irrelevant to the environment) and one of the allowed operations.

**Oracle Interventions**    To help our understanding and isolate factors to determine what the bottleneck is, we consider oracle interventions to assist policy $\pi_\theta$ by augmenting auxiliary information. First, we establish the existence of an oracle $\mathcal{O}$ that, given the prompt $x$ and the context $h_{t-1}$ is able to accurately recover the belief state of the POMDP. Then, we consider the policy $\pi$ at every step is conditioned on **oracle-augmented history** $\tilde{h}_t$:

$$a_t \sim \pi_\theta \left( \cdot \mid x, \tilde{h}_{t-1} \right),$$
$$\tilde{h}_{t-1} = \mathcal{O}^{\text{history}} \left( h_{t-1} \ \oplus \ \mathcal{O}^{\text{plan}}(x, h_{t-1}) \ \oplus \ \mathcal{O}^{\text{state}}(x, h_{t-1}) \right).$$

Generally, our oracle formulation accommodates appending ($\oplus$) a hint for the next step of an optimal plan (via $\mathcal{O}^{\text{plan}}$) and a summary of the belief state (via $\mathcal{O}^{\text{state}}$) to the history, as well as representing the context compactly by pruning the history of redundant information (via $\mathcal{O}^{\text{history}}$). We elaborate on the details of each of these in the following paragraphs.

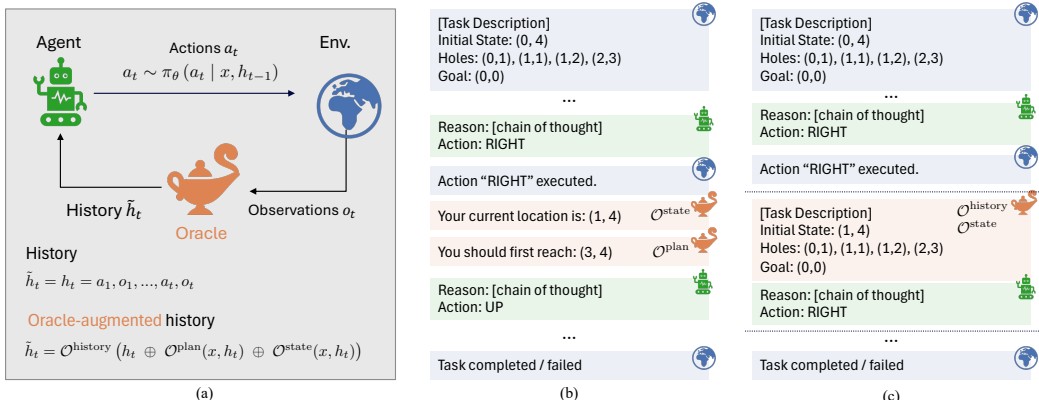

Figure 1: **Formulation.** (a) **Oracle-augmented history.** Within multi-turn tasks, we study LLM-based agents $\pi_\theta$ when additionally assisted by an oracle. We can leverage one or more oracles to modify the history $h_t$ (context for language model). (b) **GridWorld example.** In this example, the agent needs to navigate from an initial 2D location to a goal location. We can use oracle $\mathcal{O}^{\text{state}}$ to summarize the current location (instead of the model reflectively reasoning at each turn). Similarly, we can also use $\mathcal{O}^{\text{plan}}$ to hint way points to reach the goal. (c) **History pruning.** Since we consider Markov decision processes, $\mathcal{O}^{\text{history}}$ can be used to rewrite the task description such that the actions can be taken independent to previous steps.

**Planning $\mathcal{O}^{\text{plan}}$** As apparent from the POMDP formulation, solving multi-turn environments requires reasoning about many steps into the future and devising a plan towards completing the task. At every step $t$, the planning intervention $\mathcal{O}^{\text{plan}}(x, h_{t-1})$ is the description of a single-turn subtask. This subtask is designed not to require planning (no reasoning about the future steps needed). Most importantly, the action that accomplishes this subtask is one of the optimal actions in the environment at that moment.

**State Tracking $\mathcal{O}^{\text{state}}$** Solving partially-observable long-horizon tasks requires the agent to accurately track its knowledge about the hidden state of the environment at every step. This is highly challenging, since at each turn the agent needs to collect the information implicitly from its history of interactions with the environment and reason about the environment transitions. Consequently, we consider an oracle belief function that accurately summarizes the current knowledge of the agent (e.g., current location in GridWorld) in a compact form that is easy to parse.

**History Pruning $\mathcal{O}^{\text{history}}$** It is well-known that the performance of LLMs degrade as the size of the context (history $h_t$ in our case) grows. More relevant to us, existing work (Laban et al., 2025; Vodrahalli et al., 2024) highlights that the performance at the same task drops merely due to the presence of distractors. This is a common problem in multi-turn LLM agentic tasks, where the history contains overcomplete information to guide the agent towards the goal, and since the size of the context (history $h_t$) grows at each turn, makes decision making more error prone. To mitigate the influence of distractors, we consider an oracle $\mathcal{O}^{\text{history}}$ that reduces the contents of the context into a compact form. In this work, we consider the simple implementation that can be done when state tracking is present. In this case, we drop the old history $h_{t-1}$, since when the compact summary $\mathcal{O}^{\text{state}}(x, h_{t-1})$ is given, $h_{t-1}$ is not necessary.

**Step vs Task Metric $J_{\text{step}}$** In multi-turn environments, the success rate can often be low because agents are required to take the correct action at each step consistently. This challenge is compounded by the unique difficulties inherent in such environments, such as planning and state tracking. For example, even if the environment involves solving single-turn step-by-step reasoning problems, the performance can suffer due to the dependency on the chain, irrespective of the absence of planning. However, there are cases where this setup can be advantageous by offering the agent retries; for instance, in scenarios where solving a fraction of tasks is enough for success. To measure this aspect, we define an objective over each step. We call a step accurate if the action taken is optimal for that step. We refer to this metric as *step accuracy*. The distinction between step accuracy and success rate lies in how forgiving the environment is to mistakes.

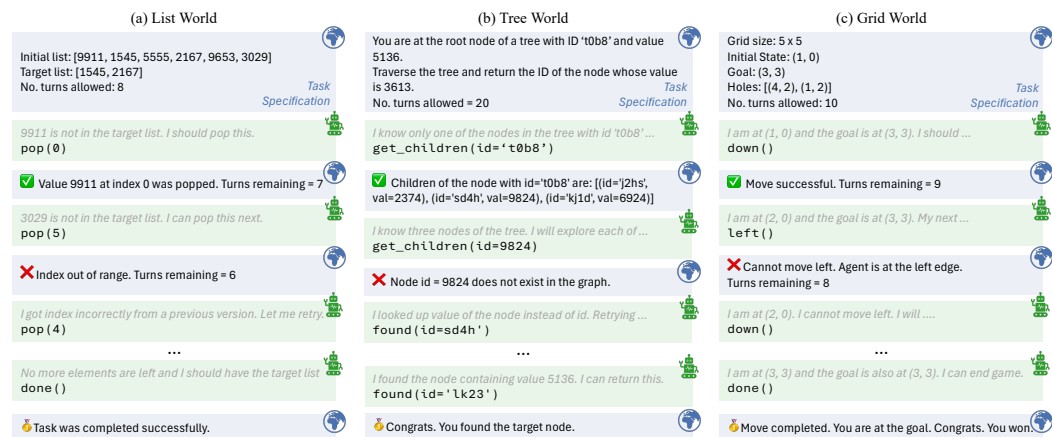

**Figure 2: Environments.** In this work, we study the influence of oracle interventions in three unique environments. In all cases, the agent reasons (shown in gray italic) and performs an action (shown in monospace), and the environment provides minimal but sufficient feedback to help the agent progress towards the goal. **(a) List World**: which requires modifying a python list using only `pop(idx)` actions; **(b) Tree World**: where the task is to iteratively search over a tree to find a specific node; and **(c) Grid World**: where the agent needs to move from an initial location to a goal location.

## 4 EXPERIMENTAL RESULTS

In this section, we first present the programmatically generated environments and tasks used in our experiments. Then, we walk through the implementation details and conclude by reporting our findings.

### 4.1 PROCEDURALLY-GENERATED MULTI-TURN ENVIRONMENTS

Our goal is to characterize bottlenecks of multi-turn agents by leveraging oracle interventions. Existing benchmarks fall short for this task, since data are predominantly human-generated and are rarely accompanied with trajectory-level annotation. Some works (e.g., Trivedi et al. (2024)) investigate marginal gains through oracle interventions, however in a very narrow scope that is admissible within the dataset. Consequently, we propose procedurally-generated multi-turn environments with the following requirements: (a) Minimal external knowledge: such that all necessary information can be specified in the prompt; (b) Simple action space: to prevent failures from constructing complex function calls; (c) Variable complexity: to enable us analyze success by varying the complexity of the task in a procedural manner; (d) Compositionality: such that tasks can be programmatically and accurately broken down into clear subproblems; (d) No data contamination: since the tasks are novel and can additionally be randomly re-generated, there is little risk from contamination; and most importantly, (e) Oracle interventions: since we know the underlying process at any instant, we can faithfully construct various flavors of oracle interventions.

**General Framework** All our environments can be cast as a Partially Observable Markov Decision Process (POMDP). Given an initial task $x$ that can be communicated verbally, the agent needs to complete the task with a finite turn budget $T_{\max} \geq mT^* + n$, where $T^*$ is the number of actions required by an optimal policy. To achieve the goal, the agent interacts with the environments using a simple set of actions (e.g., up, down). All environments have a common terminating action done, which the agent needs to invoke once it completes the objective. The environment provides minimal essential feedback (e.g., 'move successful') at each turn.

**List World** Inspired by Vodrahalli et al. (2024), we introduce ListWorld to evaluate the ability of an LLM agent to sequentially modify and track the state of an initial object. Specifically, the task of the agent is to prune an initial `input` Python list to a smaller `target` list. The agent needs to prune using a single action: `pop(index)`. The agent has to pop the elements from left to right: once an element is popped, it becomes illegal to pop the elements before it. We control the task

complexity by varying the number of elements that need to be pruned (i.e., `len(initial) - len(target)`). To complete the task successfully, the agent at each turn needs to: (a) determine current list by accounting initial list and historical actions; (b) find the candidates to prune; and (c) pop the corresponding candidate. This introduces a subtle challenge of understanding partial changes to the index-value mappings after every successful pop operation. Furthermore, the agent needs to carefully reflect before each action, since pruning an unnecessary element places the agent into an irrecoverable state leading to immediate termination.

**Tree World** In this environment, we study an agent's ability to sequentially explore and gather information at each turn. Specifically, the task is tree traversal: the agent needs to navigate from a source node to a target node to find the node containing a particular value. The nodes (except source and target) and edges are unknown to the agent. For simplicity and ease of analysis, we consider traversing from the root to a leaf node of a tree. The agent needs to traverse the tree using a single action: `get_children(node_id)`. Efficiently completing this task requires the agent to keep track of the frontier of unexplored nodes and sequentially explore them. We vary the complexity of the task by controlling the number of nodes in an $m$-ary tree. This task is partially inspired by interactive coding problems (Trivedi et al., 2024), which require an agent to navigate a new library documentation (tree topology) to find the right function to invoke.

**Grid World** We also consider a 2D grid world environment to study an agent's ability to plan and spatially navigate towards a goal. Our grid world takes the form of an $N \times N$ grid with holes, where stepping into the holes incurs an additional cost. The environment is fully observable, with each task stating the agent's start position and the goal position. The agent needs to navigate to the goal using one of four actions (`up()`, `down()`, `left()`, or `right()`) and reach the goal within a specified cost budget. To successfully complete the task, the agent needs to: (a) understand and reason spatial structure of the environment; (b) reason carefully to plan a trajectory avoiding holes; (c) keep track of progress towards the goal.

**Oracle Specifications** In all the environments above, the oracle is designed either to *augment* the information provided (e.g., by providing a plan or summarizing the belief state) or *truncate* to sufficient information necessary to achieve the goal. When we truncate the context, we signal it by rewriting the task specification and providing the current state as the initial state (e.g., current 2D location in Grid world) and discarding the previous history. A specific benefit in our environments is that we evaluate the optimal plan from *any* state that the environment is in.

### 4.2 SETUP: LLM AGENTS

In this section, we walk through the models we used for evaluation, how the prompts were designed to elicit closed-loop interactions with the environment, and also discuss evaluation metrics.

**Models** We ran all our experiments using the Qwen-3 (Yang et al., 2025) family of models as the policy model $\pi_\theta$. This family of models is appealing for our understanding multi-turn scenarios for two reasons: (i) it enables us to study performance over a range of different model sizes (we focus on 4B - 32B); and (ii) the models are already pre-trained on multi-turn interaction cycles, during the RL stage (Yang et al., 2025), and hence they are well-suited for our analysis. We report all findings by running inference in non-thinking mode (but with reasoning traces using chain-of-thought prompting) with the context length limit set to 32K and recommended sampling temperature of 0.7. We also ran preliminary experiments with thinking mode, but we found lower success rates with most failure cases due to hitting token limits. We use the ReAct (Yao et al., 2023b) framework to perform the roll-outs by interleaving reasoning traces with actions. The number of turns for which the roll-outs are performed is example dependent. Since many of our experiments are long-horizon and do not fit into the context length, we report results for such scenarios (involving contexts >32K) using YaRN (Peng et al., 2023) encoding, following the official recommendation.

**Prompting** Across all environments, we engineer environment-specific prompts to ensure best success of the pre-trained models. This helps us ensure that at evaluation time, errors can be attributed with high confidence to limitations of the model rather than prompt construction. Specifically, we: (a) use in-context example trajectories, which demonstrate task-specific reasoning and

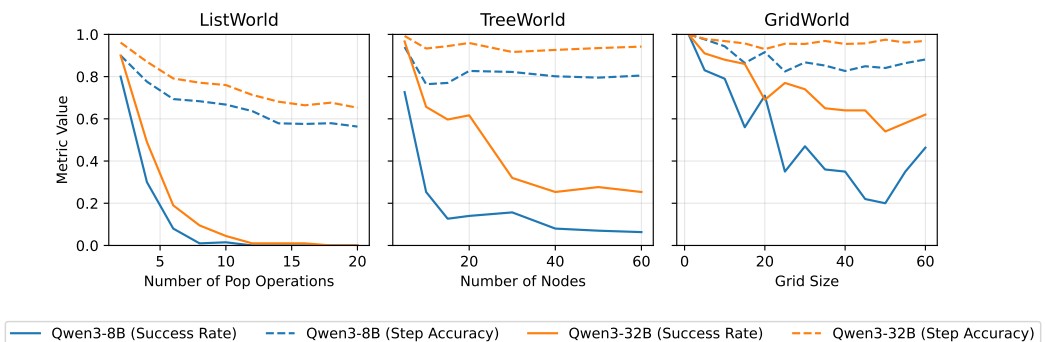

**Figure 3:** Success rate and step accuracy of Qwen3-8B and Qwen3-32B models by task horizon in ListWorld *(left)*, TreeWorld *(middle)*, and GridWorld *(right)*.

actions; (b) ensure the in-context examples reflect the information augmented by the oracles during roll-outs. We found the latter to be especially important, as models underperformed if the format of in-context examples did not appear consistent with environmental feedback during roll-out.

**Evaluation Metrics** Our primary evaluation metric is the success rate, i.e., whether the model completes the task within the specified horizon budget. In addition, in some cases we report the step accuracy. In this context, a step is "accurate", if it is optimal. Since for the environments we consider, optimal policies are generally not unique. A step is accurate if it is aligned with at least one optimal policy of the current state.

## 4.3 RESULTS AND ANALYSIS

We examine the performance of Qwen-3 models on our environments. Figure 3 shows the success rate of the Qwen3-8B and Qwen3-32B models on our three environments as a function of the task horizon. We observe that despite the strong performance on problems with a short horizon, the success rate drops drastically as we increase the horizon. This is aligned with the notorious challenging nature of long-horizon tasks. We now investigate the main driving forces behind this phenomenon with our framework.

**The dependence of task success on being correct over many steps is the main reason for low success rate in multi-turn environments.** The dashed lines in Fig. 3 show the step accuracy of the model for each value of task horizon. We can see that the step accuracy is much larger than the task success rate. Even in cases where the success rate is almost zero, the step accuracy stays above 60%. This means that even in the most challenging problem instances, the model is taking the correct action in the majority of steps. However, as the number of required steps increases, the agent becomes more likely to fail due to the occasional wrong actions it takes. In order to solve these tasks reliably, the agent needs to be almost perfect at all steps. As we have discussed, this is highly challenging in multi-turn environments due to challenges of state-tracking, planning, and growing prompt length.

To better understand the impact of each challenge of multi-turn environments, we conduct a thorough evaluation of the models across all sizes in the presence or absence of our oracle interventions. Each of these interventions removes one of the aforementioned challenges, and their impact on the agent's success rate allows us to understand the importance of each one.

Planning and state tracking interventions can be activated or deactivated independently. In cases where state-tracking intervention is active, the provided state contains sufficient information for the agent's decision-making. In these cases, we can choose to apply history pruning to reduce the content in the model's context window. Therefore, we have six possible configurations for the oracle intervention. Figure 4 provides the success rate of 4B, 8B, 14B, and 32B models in each environment for all six oracle configurations, averaged over complexity levels. For each environment, we also provide the aggregate results averaged over the four model sizes. The addition of each intervention

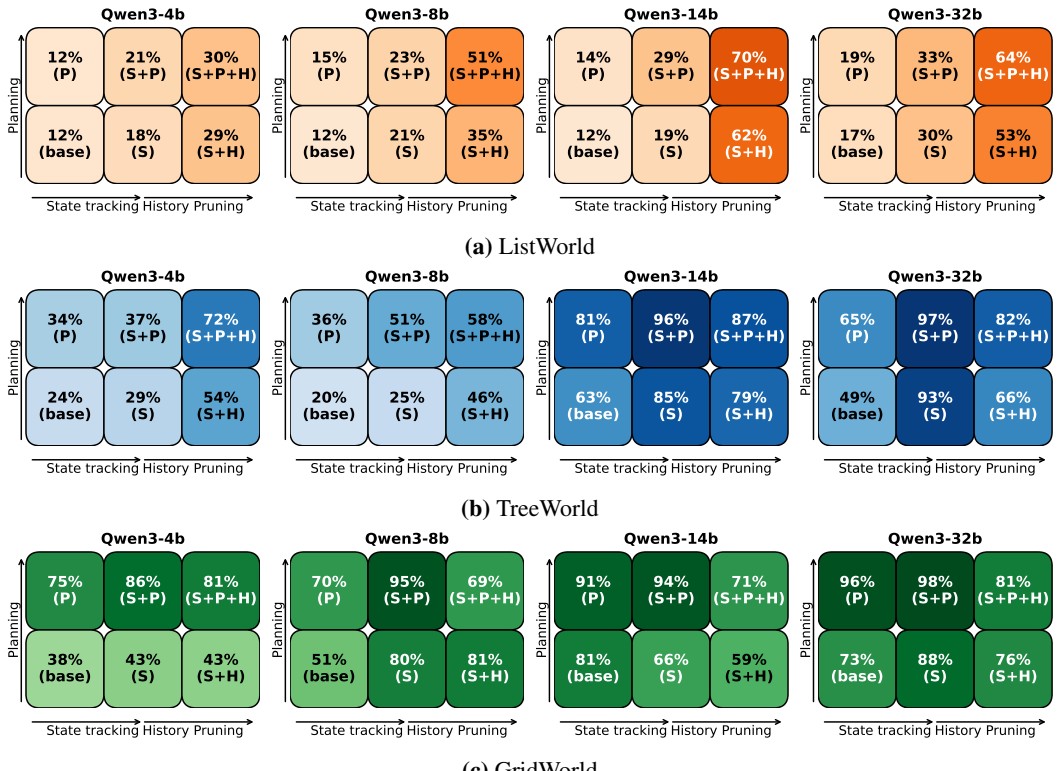

**Figure 4: Influence of oracle interventions.** Results are averaged over all horizon lengths. The labels indicate the active oracles (S: state tracking, P: planning, and H: history pruning) compared to the base model.

improves the success rate in most cases. The exact comparison of the impact of each challenge varies among the environments and model sizes.

To better understand how the bottleneck factor of models varies among different sizes, we look into the impact of each oracle intervention at that size. For a fixed benchmark, the success rate of bigger models is generally larger than the smaller models. Since the success rate is upper bounded by $100\%$, it means that the oracle interventions have less room to improve the success rate of bigger models and their impact will naturally be smaller. To enable an insightful comparison of the challenges across size categories, we utilize the programmability of our environments to adjust for this natural imbalance. We pick a longer horizon for larger models such that all models' success rates become similar. For ListWorld and TreeWorld, we challenge each model to succeed only $30\%$ of the times. In GridWorld, larger models never reach this low success rate, and we pick the complexity such that success rate becomes about $75\%$ for all model sizes. We present this comparison in Figure 5.

In the left plot of Figure 5, we provide the change in success rate due to each intervention for each model size. We average over the choice of environment and the presence/absence of other interventions. **The most drastic difference between small and larger models is in the context processing.** While the 4B and 8B models immensely benefit from removing the irrelevant parts of the context, the 14B and 32B even suffer from this removal. This indicates that for larger models, the growing context is less of an issue, and it even helps the agent. The second observation we make is that state tracking becomes more and more a bottleneck in larger models. We interpret this as other challenges are better overcome by size but state tracking remains challenging. We don't observe any specific trends in planning and find the difference between task success rate and step accuracy uniform among sizes.

Figure 5 (right) compares the impact of each intervention in different environments. We use the same method to adaptively choose the task complexity and average over the choice of model size and the presence of other interventions. **We observe that the relative severity of challenges significantly varies among environments.** In ListWorld, history pruning and choosing step accuracy introduce

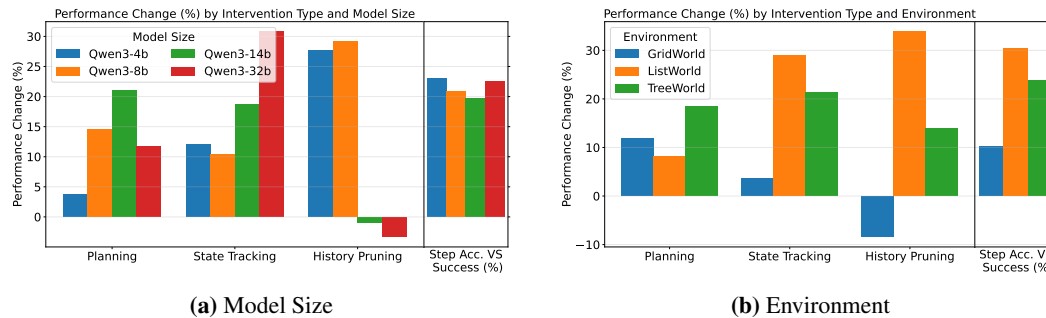

**(a)** Model Size          **(b)** Environment

Figure 5: **Performance change from interventions.** The impact of each intervention and varies depending on the model size (a) and environment (b). The relationship between step accuracy and success (%) also varies.

the largest boost in performance. This can be explained by the fact that in complex cases, the task's list is significantly long and keeping it in the recent messages helps the model to attend to it. Also, among our environments, ListWorld is the only environment that is completely irrecoverable. A single wrong action leads to task failure, hence the largest drop from step accuracy to task success rate is observed in ListWorld.

Gridworld's most demanding skill is shown to be planning, which is reasonable due to the navigation nature of the task. We find state tracking and history pruning to have a minimal benefit and even a negative impact in GridWorld. We hypothesize that this is due to the simple state update rule that can be effectively done from the context. Lastly, TreeWorld appears to be mostly demanding state tracking, perhaps due to the need to backtrack during the tree traversal. The drop from step accuracy to task success rate is also large, likely due to frequent agent's premature termination.

## 5 CONCLUSION

In this paper, we worked towards understanding the discrepancy of LLM performance between excelling at a range of complex single-turn reasoning tasks and underperforming in multi-turn closed-loop feedback-driven tasks. Our key insight was to ground the discrepancy in terms of additional skills that are required in agent-specific use cases, such as planning, learning from errors and environmental feedback, and tracking state. To enable this grounding, we proposed a simple oracle intervention framework, where the oracle complements the LLM policy by augmenting and pruning the information exposed to the agent at each turn. To support oracle interventions, we additionally propose three procedurally-generated environments (List world, Tree world, Grid world), which lets us control task complexity, and more importantly, we can at any turn estimate the set of optimal actions that can be used to guide the agent. Our findings indicate that while the skills (planning, state tracking, and history representation in our case) enable the LLM policy to generally improve, the effectiveness of each skill is also significantly influenced by the model size and the environment.

**Limitations and Future Work** This paper presents the first step towards explaining the performance degradation of capable LLMs in multi-turn long-horizon agent tasks. While we find valuable insights, many important steps remain to fully understand the performance degradation. First, we rely on prompt-based mechanisms to elicit agent-like behavior. While such mechanisms have been shown to be a strong baseline, performances are also influenced by the prompt itself, and as a result post-training is appealing to discount influence of prompt design. Second, we run our analysis on simple programmable environments and benefit from being able to accurately and efficiently construct and isolate oracle interventions. While it is insightful to study this on real-world applications, annotating oracle is often intractable or ill-defined.

## ETHICS STATEMENT

Our study focuses on leveraging LLMs in multi-turn feedback-driven scenarios. Specific to the study in this paper, we used well-established publicly-available open-weight models (however whose training is proprietary) and run our experiments on synthetic game-like programmable environments.

In this regard, we believe this is no conflict, since the environments do not contain any sensitive or personal information.

REPRODUCIBILITY STATEMENT

To ensure reproducibility, we run our analysis with open-weight models that are popularly used in the research community. Additionally, we provide relevant details and parameters in Section 4.2 and 4.3.

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

## A    ADDITIONAL EXPERIMENTS

In Figure 6, we provide the success rate of GPT-4o (Hurst et al., 2024) and Gemma 3 models (Team et al., 2025) on ListWorld for different oracle configurations. Gemma 3 results are aligned with the findings in Section 4.3 with the Qwen 3 models. Planning intervention shows minimal benefits while state tracking and history pruning significantly help the model. Both Qwen 3 and Gemma 3 models across all sizes benefit from history pruning in ListWorld, but GPT-4o, which is much larger, starts to show signs of performance degradation with history pruning. This further confirms our finding that history pruning benefits smaller models but can hurt models larger than a threshold.

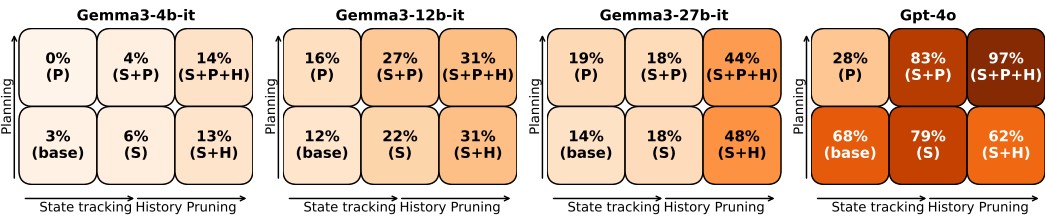

**Figure 6: Influence of oracle interventions in ListWorld.** Results are averaged over all horizon lengths. The labels indicate the active oracles (S: state tracking, P: planning, and H: history pruning) compared to the base model.

## B    PROMPTS

### B.1    PROMPT: LIST WORLD

We used the following prompt for List World experiments

```
You are an assistant designed to **modify lists** through a sequence of
 simple commands that you can execute at every turn. You will be given
an initial list and a target list. Your task is to modify the initial
list to a target list using the following functions, provided in JSON
format:
```json
{
    "pop": {
        "name": "pop",
        "type": "function",
        "description": "Removes an element from the environment's list.
        Index of the elements after the removed one will be reduced by
        1.",
        "parameters": {
            "type": "object",
            "properties": {
                "id": {
                    "type": "integer",
                    "description": "The ID of the element to remove."
                }
            },
            "required": [
                "id"
            ]
        }
    },
    "done": {
        "name": "done",
        "type": "function",
        "description": "Terminates the task. Should be called when no
        more operations are needed.",
        "parameters": {
            "type": "object",
```

```
            "properties": {}
          }
        }
      }
    }
    ```
    The initial list has all the elements of the target list in the same
    order, but contains some extra elements that need to be removed. To
    remove them, you should use the 'pop' function with the index of the
    element you want to remove. Remember that once an element is removed,
    the index of elements after it will decrease by 1. The most important
    rule is that you can **only pop the elements from left to right**. Once
     you pop an element, the elements before it (thoses with a smaller
    index) can no longer be removed, and you will get an error if you try
    to do so. Once you are done and have turned the initila list to the
    target list, you should call the 'done' function.

    You must reason step-by-step, choosing actions based on the current
    state of the list. Avoid redundant queries and aim for efficiency.
    Provide your response as a single python function call enclosed in a
    code block:
    ```python
    function_call(arg1=val1, arg2=val2, ...)
    ```

    === Starting new task ===
    Initial list: ${initial_list}
    Target list: ${target_list}
    Your task is to modify the initial list to target list by calling the '
    pop' function. Call 'done' function once you are done. Remember, **only
     pop the elements from left to right**.
```

## B.2 PROMPT: TREE WORLD

We used the following prompt for tree world:

```
SYSTEM_PROMPT = f"""You are a reasoning agent searching a tree for a
node with a specific value (which may or may not be reachable by you).
Each node has two attributes: (1) "id" (unique string) and (2) "value"
(unique integer). In each task, you are provided with a partial
information about some of the nodes in the tree. For each node included
 in this information, you are given the id and the value. For some of
the nodes the list of the node's children ids is also provided. The
format in that case is: (id=<node_id>, value=<node_value>) -> [<
child_id1>, <child_id2>, ...]. Note that an empty list indicates that
the node is a leaf and has no children. For other nodes, the children
are not given to you. In that case, you are given: \"UNKNOWN\" in place
 of the children ids.

A target value will be given to you at the beginning of each task. Your
 job is to try to find a node with this value and report its id. You
should do this using the following functions, provided in JSON format:
```json
{
    "get_children": {
        "name": "get_children",
        "type": "function",
        "description": "Returns the list of children nodes for a given
        node ID (i.e., the outgoing edges)",
        "parameters": {
            "type": "object",
            "properties": {
                "id": {
                    "type": "string",
```

```
756                        "description": "The ID of the node whose children are to
757                         be retrieved."
758                    }
759                },
760                "required": [
761                    "id"
762                ]
763            },
764            "returns": {
765                "type": "array",
766                "items": {
767                    "type": "object",
768                    "properties": {
769                        "id": {
770                            "type": "string"
771                        },
772                        "val": {
773                            "type": "integer"
774                        }
775                    },
776                    "required": [
777                        "id",
778                        "val"
779                    ]
780                },
781                "description": "List of child nodes as objects with 'id' and '
782                val'."
783            }
784        },
785        "found": {
786            "name": "found",
787            "type": "function",
788            "description": "Indicates that the node with the specified ID
789            contains the target value.",
790            "parameters": {
791                "type": "object",
792                "properties": {
793                    "id": {
794                        "type": "string",
795                        "description": "The ID of the node that contains the
796                        target value."
797                    }
798                },
799                "required": [
800                    "id"
801                ]
802            },
803            "returns": {
804                "type": "string",
805                "description": "Confirmation that the node with the given ID
806                contains the target value."
807            }
808        },
809        "unreachable": {
810            "name": "unreachable",
811            "type": "function",
812            "description": "Indicates that the node with the target value is
813            impossible to reach.",
814            "parameters": {
815                "type": "object",
816                "properties": {}
817            },
818            "returns": {
819                "type": "string",
```

```
            "description": "Confirmation that the target value was not
            possible to find in the tree."
        }
    }
}
```
You can ask for the ids and values of the children of a node by calling
 the `get_children` function with the node's id.
If you find the target node (the node with the target value), return
its id using the `found` function. After calling `found` the task will
terminate and you succeed if you have reported the correct id.
If you believe it is impossible to find the target node, call the `
unreachable` function. After calling `unreachable` the task will
terminate and you succeed if it was impossible for you to find the
target node.

You must reason step-by-step, choosing actions based on the current
state of the search. Avoid redundant queries and aim for efficiency.
Provide your response as a single python function call enclosed in a
code block:
```python
function_call(arg1=val1, arg2=val2, ...)
```

=== Starting new task ===
Your task is as follows: Find the id of the node with value **${
target_node_val}**
Once you find the target node containing this value, return its id by
calling `found` function. If you think it is impossible to find this
node, call the `unreachable` function.
Provide all responses as a single python function call enclosed in a
code block.
```

## B.3 PROMPT: GRID WORLD

We used the prompt below for grid world:

```
You are an intelligent agent playing a grid world navigation game. Your
 goal is to move from the given start position to the goal position
using the fewest possible moves. The game board is a 2D grid with the
following properties:

- The top-left corner is coordinate (0, 0), and the bottom-right corner
 is (size-1, size-1).
- You will be given:
    * The size of the board (N x N)
    * Your starting position (row_index, column_index)
    * The goal position (row_index, column_index)
    * A list of hole positions (each a coordinate)
    * The maximum number of moves allowed
- You can move using these actions: `up()`, `down()`, `left()`, `right
()`
- *Only* if you have reached the goal, call `done()` to terminate the
game. Once you terminate the game, you are not allowed any more moves.
- You can reason, but always end by specifying a single action within
triple fenced blocks. Example
```python
up()
```
or
```python
done()
```
- Each move costs **1 move**.
```

```
- If you move into a hole, you incur a **penalty of 3 additional moves
** (because it is hard to get out of a hole).
- You must stay within the grid boundaries.
- Your objective: **Reach the goal in as few moves as possible without
exceeding the maximum allowed moves.**
- After each move, you will receive the updated position and remaining
moves.
- In the triple fenced blocks, do not write anything except the next
action in the required format.

=== Your Task ===
The grid world game is set up as follows:
- Board size: ${size} x ${size}
- Start position: ${start}
- Goal position: ${goal}
- Holes at: ${holes}
- Your move budget is: ${max_moves}

Your task: Navigate from the start to the goal using the fewest moves
possible. Remember:
- You can move using the following actions: `up()`, `down()`, `left()`,
 `right()`
- If you reached the goal, terminate by performing action `done()`
- Each action must be in a triple-fenced Python code block, like:
```python
right()
```
- Avoid holes if possible, as they cost extra moves.
- Do not exceed the maximum allowed moves.

Begin your first move now.
"""
```

