# OpenReview forum: "LUMINA: Long-horizon Understanding for Multi-turn Interactive Agents"
_ICLR.cc/2026/Conference — ICLR 2026 Conference Withdrawn Submission_

### Official Review · Reviewer_YWjF · 2025-10-30

**Soundness:** 3
**Presentation:** 2
**Contribution:** 2
**Rating:** 4
**Confidence:** 3

**Summary:**

To study how agents perform in long-horizon tasks given a specific oracle skill, the authors construct three text-based worlds. They also focus on long-horizon tasks for agents that can be modeled as Partially-observable Markov Decision Processing, where the oracle intervention is that  agent can recover the belief state of the POMDP accurately under such intervention. They have found out that while the skills can improve LLM's policies, the effectiveness of each skill is influenced by the model size and environment.

**Strengths:**

1. Construction of three worlds to enable oracle skill control contributes to the research community. The worlds enable faithfully constructed oracle skills to study the behavior and performance of LLM agents, which is helpful to understand what affects LLM's performance.
2. The finding about LLMs excel at each step but performs relatively poorly in the entire horizon is interesting.

**Weaknesses:**

1. I would like to see stronger models' performance, like Qwen3-235B you have mentioned in the abstract, and also GPT-4o, maybe GPT-5. I would also like to see how o3 or o4-mini models performs. I am concerned about those tasks maybe only hard enough for small open source models (Qwen3-4b can get 86% with state tracking and planning in grid world). If this is the case, those worlds are still useful but limited.
2. The oracle formulation accommodates hints, planning, state tracking and history pruning. Those are reasonable and common considerations. However, Since hint, planning, history pruning are somehow "common practice" to augment LLMs in different tasks,  I would like to see some oracle interventions that is tailored to POMDP specially other than state tracking.

**Questions:**

1. Planning, State Tracking and History Pruning are common skills. Any other skills that can be considered? (Connection to Weakness 1)
2. Would you mind providing results of Qwen3-235B/GPT-4o and o3/o4-mini/Deepseek R1 on three worlds?
3. Can we view reasoning models (e.g. o3) as one LLM agent equipped with $O^{\text{plan}}$?

---

> ### Author Response · Authors · 2025-11-26
>
> We thank the reviewer for their review and valuable feedback.
>
>
> **(Weakness 1, Q2) I would like to see stronger models' performance**
>
> Thanks for the suggestion.
> We want to highlight that as shown in Figure 3, the difficulty our environement can be easily adjusted by a single parameter that decides the task horizon. By increasing the parameter we can reliably challenge larger models. We now add results of GPT-4o and also the Gemma 3 series of models on ListWorld in Appendix A. Averaged over all complexities (choice number of number of pop operations), GPT-4o has a success rate of 68%. For 20 pop operations, this falls to 48%.
>
> **(Weakness 2, Q1) hint, planning, history pruning are common practise ... like to see other interventions**
>
> This is indeed an interesting direction to expand our analysis. At the moment, we are not aware of other POMDP-specific capability. Our chosen capabilities are natural steps that simplify the most complicated setting in sequential decision-making (history-dependent policy in a POMDP) towards the simplest one (bandits). State intervention ($O^\mathrm{state}$) turns the POMDP with history-dependent policy into an MDP with a history-dependent policy. History pruning ($O^\mathrm{history}$) turns this into an MDP with a Markov policy. Lastly, planning intervention ($O^\mathrm{plan}$) leads to a bandit problem.
>
> **(Q3) Can we view reasoning models (e.g. o3) as one LLM agent equipped with $O^{\text{plan}}$?**
>
> Not quite. The long chain-of-thoght (CoT) generated by these models is beyond just planning (as defined in our paper). We expect the CoT to also include thoughts about state tracking, formation of the function call, reiterating environment description, long-term plan, etc. Our $O^\mathrm{plan}$ only includes a subtask for the next immediate action and is only one possible thought a reasoning model can generate.

---

### Official Review · Reviewer_M8ES · 2025-10-30

**Soundness:** 2
**Presentation:** 2
**Contribution:** 2
**Rating:** 2
**Confidence:** 3

**Summary:**

This paper investigates the performance gap between LLMs on single-turn tasks and their underperformance on multi-turn tasks. The authors attribute this gap to additional skills required in multi-turn settings, including long-context reasoning, planning and decision making, and state tracking. Experiments conducted in controlled environments shows that compounding errors constitute a primary source of failure.

**Strengths:**

This paper provides a clear analysis of compounding errors in long-horizon tasks, showing how small step-wise mistakes accumulate to reduce overall success. It further disentangles specific agentic skills and introduces controlled environments to assess their individual contributions. Experiments across multiple skill combinations and model scales reveal that larger models can leverage longer contextual dependencies more effectively.

**Weaknesses:**

1.  The proposed environments are symbolic and fully rule-defined, omitting key challenges of real-world tasks such as parsing unstructured feedback. It is therefore unclear whether the identified bottlenecks generalize to real-world tasks.

2. Some of the findings have been reported in other works, which may limit the novelty of the results. For instance, recent studies on memory-augmented and planning-based agents have shown that these components can substantially influence performance.

3. All experiments are conducted with the Qwen-3 model family, limiting the findings to Qwen-3’s scaling behavior. Other model families, such as Llama or Mistral, may exhibit different scaling dynamics and bottleneck characteristics.

**Questions:**

1. The study reports binary success rates as the primary metric. Could the authors provide additional measures, such as the ratio of actual-to-optimal path length, to better assess the impact of agent skills?

2. The results suggest that larger models can leverage longer contextual dependencies more effectively than smaller models. Could the authors evaluate how interaction history length affects performance across model scales, as shorter histories may benefit smaller models while excessively long histories could reduce performance for larger models?

3. The paper notes that performance declines when in-context examples are not aligned with oracle feedback, suggesting sensitivity to prompt design. Ablation studies on prompt wording or structure could clarify this.

4. The paper mentions that $O^{plan}$ provides a description of a single-turn subtask, but its precise nature is unclear. Could the authors clarify whether this corresponds to step-by-step guidance or a lower-level specification of the immediate action, and how each type affects model performance?

---

> ### Author Response · Authors · 2025-11-26
> **Response to Reviewer M8ES (part 1)**
>
> We thank the reviewer for their thorough review and valuable feedback. We would like to address the concerns raised in the review.
>
> **(Weakness 1) Proposed environments are symbolic and fully rule-defined ... avoids challenges of real-world tasks**
>
> The reviewer is right that in this paper, we have avoided some challenges of AI agents in real-world tasks, such as the unstructured feedback by the user. We want to emphasize that this was indeed an intentional choice for two main reasons.
>
> First, the presence of task-specification challenges, e.g. unstructured instructions, obscures the study of the fundamental agentic capabilities that are the focus of this paper. For instance, the planning capability of the model cannot be measured without ensuring the goal is clearly communicated to the model. Due to this prohibitive impact, targeted studies are needed that avoid these realistic challenges in order to study the more fundamental capabilities. We aim to address this gap in the literature and focus on the insights that have remained hidden to the past work, which only focus on the most realistic scenarios. Our rule-based environments allow us to provide in-context examples with the exact same feedback structure as the real task and avoid any errors due to task misspecification.
>
> Second, the programmatic nature of the environments has key advantages. The difficulty and the horizon of the task can be easily adjusted in our environments, which makes our framework suitable to study the agentic capabilities of a wide range model sizes, even the more capable models that will be introduced in the future. This design choice also ensures accurate oracle interventions.
>
> **(Weakness 2) Some findings (e.g., memory-augmentation) already report improved performance ... limits novelty of results**
>
> We want to highlight that while some skills considered in this paper have been studied individually, little is known about their relative importance in comparison. Our contribution is beyond establishing the skills required for AI agents and importantly, also covers the comparison of their importance towards identifying the major bottleneck in each model size. These insights are highly valuable for practitioners to direct the time and energy towards improving the critical aspects of the agents. For example, we have shown developing context summarization techniques is more important than planning guidance for agents that are based on a small model. Such understandings are only possible due to our novel framework to measure the impact of the skills in a unified manner and is absent in the previous work that only study one aspect.
>
> **(Weakness 3) Only Qwen3 results reported**
>
> We now complement previous Qwen3 (4B, 8B, 14B, 32B) results with additional results on Gemma 3 models (4B, 12B, 27B) and GPT-4o on Listworld environment.
> The results are aligned with the previous findings. Planning intervention shows minimal benefits while state tracking and history pruning significantly help the model. Both Qwen 3 and Gemma 3 models across all sizes benefited from history pruning in ListWorld, but GPT-4o, which is much larger, starts to show signs of performance degradation with history pruning. This further confirms our finding that history pruning benefits smaller models but can hurt models larger than a threshold.
>
> **(Q1) Only reports binary success rate ... to report actual-to-optimal path length**
>
> We appreciate the reviewer's suggestion and will try our best to provide additional results with partial success evaluations in the next version. We expect the results with such metrics to be qualitatively aligned with the current results of the paper.
>
> **(Q2) Could the authors evaluate how interaction history length affects performance across model scales?**
>
> Indeed, longer interaction lengths reduce the performance of the models. We already visualize this effect and compare the trends for small and large models in Figure 3 of the paper. We are happy to provide additional results if any other specific result can better address the concerns.
>
> **(Q3) sensitivity to prompt design. Ablation studies on prompt wording or structure could clarify this.**
>
> Our prompts used in the paper were designed based on preliminary exploration and experiments (e.g., number of in-context examples, addition/removal of oracles) in a small-scale setting.
> We are glad to include more rigorous analysis in the next version.

---

> > ### Author Response · Authors · 2025-11-26
> > **Response to Reviewer M8ES (part 2)**
> >
> > **(Q4) What does $O^{\text{plan}}$ do precisely?**
> >
> > $O^\mathrm{plan}$ adds a hint message to the chat history observable by the agent (as depicted in Figures 1b and 1c).
> > The content of the hint message varies by environment and is defined as:
> >   - List world: If the current list is the target list, oracle hints to terminate. Otherwise, the oracle hints on the next valid value to pop.
> >   - Tree world: The hint is one of: (a) the history already contains the target value; (b) The tree is fully explored by the history and does not contain the target value; or (c) agent needs to continue exploration to find the target value.
> >   - Grid world: If the agent is already at the goal location, the oracle hints to terminate. Otherwise, it hints at which adjacent location (along an optimal route) the agent needs to move to.

---

### Official Review · Reviewer_WkEG · 2025-11-01

**Soundness:** 4
**Presentation:** 4
**Contribution:** 3
**Rating:** 6
**Confidence:** 3

**Summary:**

LUMINA offers a principled way to analyze why LLM agents fail in complex, multi-step interactions — by decomposing agent behavior into modular skills and testing them systematically. To address this, the authors introduce LUMINA, a controlled evaluation framework using oracle counterfactual interventions: They design procedurally generated game-like environments where agent goals and task complexity can be precisely controlled. The oracle can intervene to provide specific “skills” (e.g., planning, exploration, state tracking), allowing researchers to test how each skill contributes to final performance.

**Strengths:**

The paper identifies a crucial and underexplored limitation in current LLM-based agents—their inability to maintain robust long-horizon reasoning across multiple turns. The motivation is well-grounded in empirical evidence (e.g., low success rates despite high per-step accuracy), and the authors effectively position long-horizon understanding as a distinct capability beyond standard reasoning or planning.

The introduction of an oracle counterfactual intervention framework is a major methodological strength. By isolating specific skills (e.g., planning, tracking belief state, context reformulation) and testing their contribution to success, the paper provides a systematic, interpretable way to analyze agentic competence—something rarely achieved in prior multi-turn benchmarks that often rely on end-to-end success metrics.

**Weaknesses:**

While the proposed environments (ListWorld, TreeWorld, GridWorld) are carefully controlled and effective for isolating individual skills, they remain relatively synthetic and detached from widely adopted agentic benchmarks such as ScienceWorld, OSWorld, or TravelPlanner. As a result, the paper provides valuable mechanistic insight but lacks direct evidence that the identified skill bottlenecks generalize to real-world multi-turn tasks. This limitation weakens the practical applicability and external validity of the findings.

The paper’s focus is primarily diagnostic rather than improvement-oriented. Although the oracle intervention analysis yields interpretive insights, it does not translate into a clear enhancement of actual agent performance. The study stops short of proposing or validating concrete training or inference strategies that could operationalize these insights to improve long-horizon reasoning capabilities. Thus, the contribution remains more analytical than actionable.

**Questions:**

as weakness

---

> ### Author Response · Authors · 2025-11-26
>
> **(Weakness 1) Main concern is synthetic experimental setting ... more interesting to run analysis on real-world scenarios**
>
> We agree with the reviewer that expanding our analysis to real-world scenarios is a natural next step to further understand the bottlenecks in AI agents. Our framework of oracle intervention is compatible with the real-world tasks. However, they impose technical challenges. The oracle interventions require a ground-truth solution to assist the model with a compact message, which might be impossible or labor-intensive to obtain in many realistic tasks. Also, unstructured feedback can create misunderstandings for the model that dominate the failures due to core capabilities. We hope to overcome these challenges in future work.
>
> **(Weakness 2) Work feels incomplete ... hoping to see concrete training and inference strategies**
>
> We want to highlight that assisting the model with programmed software is a common practice in design of AI agents to improve their performance [1]. Our work can be viewed beyond diagnosing shortcomings of models and seen as an investigation of this improvement technique.
>
> [1] Marreed, S.; Oved, A.; Yaeli, A.; Shlomov, S.; Levy, I.; Akrabi, O.; Sela, A.; Adi, A.; Mashkif, N. Towards Enterprise-Ready Computer Using Generalist Agent. arXiv July 9, 2025. https://doi.org/10.48550/arXiv.2503.01861.

---

### Official Review · Reviewer_aTbk · 2025-11-01

**Soundness:** 2
**Presentation:** 3
**Contribution:** 2
**Rating:** 4
**Confidence:** 4

**Summary:**

This studies what specific skills make LLMs effective as multi-turn agents. The authors build an oracle intervention framework that can add idealized capabilities to an LLM policy during rollouts, such as planning hints, belief-state summaries, and context pruning. They introduce 3 generated environments—ListWorld (iterative list edits), TreeWorld (graph traversal), and GridWorld (2D navigation)—with controllable complexity and computable optimal actions.

**Strengths:**

Three procedurally generated environments enable controllable complexity.  They are designed with simple action spaces and trajectory-level annotations, supporting accurate measurement of optimal actions.

**Weaknesses:**

The idea of using oracle-based counterfactual interventions to dissect agent capabilities is interesting.
However, I have some concerns. The three oracle modules are treated as independent switches, but they interact tightly. Oplan converts the decision into a one-step optimal subtask, inherently reducing the need for state inference or history recall. Ostate summarization may already encode most of the historical trajectory.
Also  the simplification of Ohistory as truncate earlier steps is questionable, making it unclear whether the observed effects are from history or from the artificial deletion of essential cues.

**Questions:**

-  The abstract claims that “LLMs perform well on mathematics and code generation” are too broad; these capabilities were acquired after domain-specific fine-tuning.
- The reported 44.5% accuracy of Qwen3-235B on the BFCLv3 multi-turn benchmark lacks citation or reproducibility details.
- The notation alternates between (O_{state}), (O_{belief}), and (O_{context}). Consistent naming would improve readability.
- Several grammatical errors should be fixed, e.g., *“a oracle” → “an oracle”*, *“recieves” → “receives”* (line 139).

---

> ### Author Response · Authors · 2025-11-26
>
> We thank the reviewer for their thorough review and valuable feedback. We would like to address the concerns raised in the review.
>
> **Oracle modules are treated as independent switches, but they tightly interact**
>
> This is a great observation.
> Indeed, the oracle interventions may partially affect each other, and we do not assume otherwise in our paper.
> Hence, we run all possible combinations of interventions and measure the impact of an intervention for all possible choices of other interventions. On top of the general insights we draw by aggregating the setups, we provide fine-grained evaluation in Figure 4. In Figure 4, the comparison any two adjacent cells is the impact of the intervention in one individual setup.
>
> We do not see this nuance of the measurements as a weakness and even consider it a strength. We consider the end goal of this work to be guiding the design of AI agents where a model is assisted with programmatic signals. The benefit of each signal varies based on the presence of other signals. This inherent phenomenon is reflected and measured in our results, which only makes them more informative towards the downstream goal of this study.
>
> **"LLMs perform well on mathematics and code generation" are too broad**
>
> Thanks for pointing out. We slightly revised the wording.
>
> **Reported 44.5\% accuracy on BFCLv3 multi-turn lacks citation and reproducibility details**
>
> We substituted this statement with a similar one with a published reference instead of an online leaderboard (details and citation in the main text).
>
> **Notation alternates ... add consistent naming ... Several grammatical errors**
>
> We apologize for inconsistent terminology. We revised the paper and now make the terms consistent. We made another pass to fix the grammatical errors.

---

### Official Review · Reviewer_hdG4 · 2025-11-02

**Soundness:** 3
**Presentation:** 3
**Contribution:** 3
**Rating:** 6
**Confidence:** 4

**Summary:**

Language models have demonstrated strong performance across a wide range of tasks, such as mathematical reasoning and coding, which are fundamental to solving more general goal-oriented and feedback-driven agentic problems. However, such agentic problems require a diverse set of capabilities, including long-context reasoning, planning and decision making, and efficient exploration. Even large frontier models still underperform on this family of tasks, particularly those involving long-horizon understanding. For example, Qwen3-235B achieves only 44.5% accuracy on the BFCLv3 multi-turn benchmark. This paper aims to investigate which specific skills are essential for effectively solving multi-turn problems. To this end, it introduces an oracle intervention framework that evaluates the importance of different skills by posing counterfactual questions. The study finds that while most interventions, such as improving planning, are generally beneficial, the utility of certain interventions depends on the nuances of the benchmark, for example, the ability to accurately track state while iteratively modifying Python lists.

**Strengths:**

Originality: The paper designs three procedurally generated multi-turn environments, which facilitate the study of which skills have the greatest impact on agent capability.

Significance: Analyzing which skills, or combinations of skills, constitute the main bottlenecks to advancing capable multi-turn agents is highly meaningful, as it provides guidance for targeted improvements.

Clarity: The paper is clearly written.

**Weaknesses:**

Quality: The capability improvements observed in simulation environments may not necessarily transfer to real-world settings.

Significance: Can the conclusions drawn from simulation environments be applied to benchmarks in real-world scenarios?

**Questions:**

1. How are the different agent skills defined and categorized? Why does the paper focus only on the three skills: planning, state tracking, and history planning?
2. Are ( $O^{state}$ ) and ( $O^{belief}$ ) referring to the same concept in line 158?
3. In the experiments comparing the impact of different skills on performance, the trajectories are multi-step. Is the specific skill intervention applied at every step of the trajectory?

---

> ### Author Response · Authors · 2025-11-26
>
> We thank the reviewer for their review and valuable feedback. We would like to address the concerns raised in the review.
>
> **Can the conclusions drawn from simulation environments be applied to benchmarks in real-world scenarios?**
>
> This is indeed a very important yet hard-to-answer question. Real-world scenarios introduce two main challenges towards our oracle intervention framework, which makes it difficult to test our findings in them. First, the oracle intervention are much more difficult, if not impossible, to implement in real-world scenarios. The planning interventions require access to the ground truth next optimal action in every situation the model may end up in. Also, the state and history pruning require a compact comprehensive description of the state. These may require manual annotations or be inaccessible. Second, in the real-world scenarios the description of tasks (and subtasks required in planning intervention) become increasingly harder to specify. It has been shown that misspecification is a major challenge in real-world agentic tasks. This makes the performance measurements under oracle interventions an inaccurate measurement of the fundamental capability under study as other factors become more significant.
>
> **(Q1) How are skills chosen? Why only three skills (planning, state tracking, and history pruning)**
>
> Skills are chosen to simplify complex sequential decision-making setup (history-dependent policy in a POMDP) to the simple one (bandits). State intervention ($O^\mathrm{state}$) turns the POMDP with history-dependent policy into an MDP with a history-dependent policy. History pruning ($O^\mathrm{history}$) turns this into an MDP with a Markov policy. Lastly, planning intervention ($O^\mathrm{plan}$) leads to a bandit problem. Therefore, we believe these are the core capabilities that a model for agentic tasks should possess.
>
> **(Q2) Are ( $O^{state}$ ) and ( $O^{belief}$ ) referring to the same concept in line 158?**
>
> Yes. Thanks for pointing out the typo. We fixed it in the revision.
>
> **(Q3) Are skill intervention applied at every step of the trajectory?**
>
> Yes.

---

### Note · Authors · 2026-01-06

I have read and agree with the venue's withdrawal policy on behalf of myself and my co-authors.